# No Association between ABCB1 G2677T/A or C3435T Polymorphisms and Survival of Breast Cancer Patients—A 10-Year Follow-Up Study in the Polish Population

**DOI:** 10.3390/genes13050729

**Published:** 2022-04-21

**Authors:** Ewa Totoń, Barbara Jacczak, Wojciech Barczak, Paweł Jagielski, Robert Gryczka, Hanna Hołysz, Sylwia Grodecka-Gazdecka, Błażej Rubiś

**Affiliations:** 1Department of Clinical Chemistry and Molecular Diagnostics, Poznan University of Medical Sciences, Rokietnicka 3, 60-806 Poznan, Poland; etoton@ump.edu.pl (E.T.); barbarajacczak07@gmail.com (B.J.); hannak@ump.edu.pl (H.H.); 2Radiobiology Lab, The Greater Poland Cancer Centre, 61-866 Poznan, Poland; barczak.woj@gmail.com; 3Department of Nutrition and Drug Research, Faculty of Health Science, Jagiellonian University Medical College, Skawinska 8, 31-066 Krakow, Poland; paweljan.jagielski@uj.edu.pl; 4Department of Oncology, Poznan University of Medical Sciences, 60-569 Poznan, Poland; rgryczka@ump.edu.pl (R.G.); sylwiagg@ump.edu.pl (S.G.-G.)

**Keywords:** breast cancer risk, ABCB1, polymorphism, glycoprotein P, drug resistance, hormone receptors

## Abstract

Many intensive studies are devoted to identifying novel cancer diagnostics or therapy strategies that would boost cancer therapy efficacy and recovery rates. Importantly, polymorphisms in the genes coding for ABC family proteins were considered good candidates for cancer development risk or cancer drug resistance markers. For this reason, we decided to assess the contribution of ABCB1’s most common variants (i.e., G2677T/A in exon 21/rs2032582 and C3435T in exon 26/rs1045642) to the cancer therapy response in breast cancer patients. A 10-year follow-up analysis of 157 breast cancer patients was performed. Clinical assessment, ABCB1 polymorphism status, estrogen/progesterone/human epidermal receptors status, and other characteristics were compared according to the follow-up status using the Chi-square statistic. For the analysis of overall survival curves in TCGA breast cancer patients, the Xena browser was used. We show that neither 2677 nor 3435 polymorphisms contributed to the survival of breast cancer patients. Interestingly, but not surprisingly, estrogen and progesterone receptors status were good prognostic factors and positively correlated with a disease-free survival for up to 10 years. To summarize, *ABCB1* polymorphisms status may be one of the numerous factors that affect cancer development. However, they may not be the critical ones when it comes to risk or recovery assessment. Consequently, they may not be treated as reliable prognostic or predictive markers in breast cancer patients’ evaluation, which supports the previous findings and current knowledge.

## 1. Introduction

### 1.1. Cancer as a Therapy Target

Many efforts have been made to study potential diagnostic or prognostic markers to enable early diagnosis and higher recovery rates. In particular, pharmacogenomics and pharmacogenetics significantly contribute to this promising trend. Consequently, the identification of new diagnostic and prognostic markers in molecular medicine of cancer is ongoing. The main strategies in this research are case–control studies, meta-analysis, genome-wide sequencing, and follow-up studies. However, it is not easy to find a qualitative and unequivocal marker that might be used as a diagnostic, therapeutic, or predictive marker because cancer is derived from other host cells. In turn, this makes cancer cells very difficult to distinguish and target without serious side effects. Consequently, histopathological evaluation of the tumor, receptor status, and individual properties of the patient contributes to the adjusted surgical procedures, chemotherapy, radiotherapy, and combined hormonal treatment.

Breast cancer is one of the most heterogeneous and diverse tumors with the highest genetic variations and various receptors statuses [1]. Moreover, each receptor demonstrates varied prognostic and predictive values. For example, estrogen receptor (ER)-positive breast cancers comprise a significant majority of all invasive breast cancers (over 70%; [2]). Noteworthy, the vast majority of ER-positive patients also express progesterone receptors (PgR). ER- or PgR-positive patients are generally highly responsive to endocrine treatment and are associated with better treatment response and more prolonged survival [3,4]. In this case, a common approach is using selective receptors modulators, e.g., Tamoxifen [5].

The third of best characterized and well-known receptors, human epidermal growth factor receptor 2 (HER2), is overexpressed in approximately 20% of all primary breast cancers and is associated with a decreased overall survival [6]. Although HER2 overexpression is associated with faster cancer growth and a worse prognosis, a therapy based on monoclonal antibodies (e.g., Trastuzumab, Herceptin) shows some beneficial outcomes [7].

In particular, Triple-negative breast cancer (ER-, PgR-, HER2-) is perceived as the most aggressive and shows the worst overall survival rate [8]. Most treatments cannot be applied due to the absence of hormone receptors. Thus, besides surgery, adjuvant or neoadjuvant chemotherapy or radiotherapy are primarily used [9].

### 1.2. Drug Resistance

Statistical analyses show that the leading cause of cancer therapy failure is resistance to therapeutic agents [10]. It can be driven by numerous mechanisms that include multi-drug resistance, cell death inhibition (apoptosis suppression), altered drug metabolism, epigenetic modulation, changes in the drug targets, enhanced DNA repair and genes amplification [11]. Consequently, the development of resistance limits the list of available drugs [12]. Resistance of cancer cells to therapy must be considered in the context of tumor heterogeneity (including mutations, gene amplifications, deletions, chromosomal rearrangements, transposition of the genetic elements, translocations and microRNA alteration), tumor microenvironment (including cytokines and growth factors) as well as cancer stem cells that show exclusive resistance to drugs and toxins through the expression of drug efflux transporters, an active DNA-repair capacity and a resistance to apoptosis, vascular niche, dormancy, hypoxic stability and the enhanced activity of repair enzymes [13]. Some of the key players in drug resistance development are ATP Binding Cassette (ABC) family proteins. Thus, the phenotypic assessment of the protein or identification of polymorphisms in individual ABC family genes seems to constitute a promising strategy in cancer prediction, fighting, and prevention. One of the essential ABC family proteins is ABCB1, which is responsible for drug pharmacodynamics, and according to literature data, it may contribute to increased cancer risk and/or resistance of cancer to drugs [14]. This 170 kDa transmembrane protein functions as an ATP-dependent efflux pump that exports exogenous and endogenous substrates out of target/cancer cells [15]. ABCB1 is expressed in nonmalignant tissues (including the intestine and the blood–brain barrier) and malignant tissues, e.g., breast cancer and ovarian cancer tissues [16]. It is highly polymorphic, and there are at least three clinically relevant single nucleotide polymorphisms (SNPs) that are intensively studied in the context of cancer development and resistance to therapy, i.e., 1236C > T, 2677G > T/A, and 3435C > T. They are associated with altered messenger RNA stability, protein folding, drug pharmacodynamics and pharmacokinetics disturbance [17].

The most interesting SNPs in Caucasian women are ABCB1 G2677T/A in exon 21 (rs2032582) and ABCB1 C3435T in exon 26 (rs1045642) since they were shown to correlate with lower ABCB1 protein levels [18] and, consequently, with lower cellular clearance of paclitaxel [19]. These two polymorphisms have been studied in various malignancies and are associated with the risk for endometrial, breast, gastric, and colon cancer and leukemia [18]. Moreover, these polymorphisms were reported to be associated with a better response to preoperative chemotherapy and a better prognosis for breast cancer [20], esophageal cancer [21], acute myeloid leukemia [22] and multiple myeloma [23]. However, there is no consistency in the possibility of using ABCB1 polymorphisms as reliable breast cancer risk associated factors or chemotherapy response predictors. In our previous studies [24], we observed no significant differences in the studied polymorphism frequencies between control subjects and breast cancer patients. Although these polymorphisms are possibly good candidates in other tissue cancer types, we suggest that the studied polymorphisms might not be effective predictive factors in breast cancer risk or development in Caucasians. However, for a deeper understanding of the role of ABCB1 polymorphisms in breast cancer therapy response, we performed the survival assessment in a 10-year follow-up of patients.

## 2. Materials and Methods

### 2.1. Patients’ Characteristics

The current study is based on the information from the records of breast cancer patients (postmenopausal women) diagnosed with breast cancer at various stages based on WHO criteria. All patients were recruited from the Department of Oncological Surgery at the Poznan University of Medical Sciences, Poland. The patients were Caucasians from the same region of Poland (Great Poland). Tumor grading and staging were performed according to the Nottingham modification of the Bloom and Richardson Grading System and Tumor Nodus Metastases staging system [24], respectively (Table 1). All participating individuals provided a written informed agreement. The study protocol was approved by the Ethics Committee of the Poznan University of Medical Sciences (309/10), and the participating individuals provided written consent.

The original study’s results describing the frequency of ABCB1 gene polymorphisms in the polish population were published by authors in 2012 [24]. Next, the follow-up assessment was performed for up to 10 years (with 3, 5, 7, and 10-year time intervals). Briefly, a group of 157 women at various breast cancer stages (based on WHO criteria) were enrolled in the study. Polymorphism verification was performed using genomic DNA extracted from peripheral blood leukocytes using a DNA isolation kit (A&A Biotechnology, Gdynia, Poland). The polymorphisms analysis was performed by PCR (as previously described [24]) using specific primers followed by restriction analysis (C3435T, Sau3AI enzyme) or allele-specific amplification (G2677T/A). In both cases, a random 10% of all the samples were verified by sequencing using the ABI Prism 3130XL (Applied Biosystems, Darmstadt, Germany). Importantly, the number of patients gradually decreased over time (relative to the study from 2012) due to distant relocation and the loss of contact with the patients, or the parameter was not found in the database.

### 2.2. Histological Assessment

The tumors were typed according to the WHO criteria. The histological grade was determined according to the Nottingham Grading System (Elston–Ellis modification of the Scarff–Bloom–Richardson grading system) recommended by the WHO, AJCC and the Royal College of Pathologists.

#### 2.2.1. Preparation of Histological Material

Tissue samples were fixed in 10% buffered formalin at pH 7.4 and placed into a tissue processor. The samples were later embedded in paraffin wax with a melting point of 58 °C according to standard histopathological methods. The appropriately marked paraffin blocks obtained in this manner were subsequently sectioned on a microtome at a thickness of 4–5 μm. The sections were mounted onto adhesive microscope slides and incubated for 2 h at 58 °C.

#### 2.2.2. Immunoperoxidase Methods

For the purpose of demonstrating HER2, the Dako HerceptTest kit was applied. Estrogen receptors and progesterone receptors were demonstrated with Dako monoclonal antibodies. For the demonstration of markers, RTU antibody solutions were applied as follows:Mouse monoclonal antibodies raised against human α estrogen receptors (Dako Denmark A/S, Clone 1D5, No: M 7047);Mouse monoclonal antibodies raised against human progesterone receptors (Dako Denmark A/S, Clone PgR 636, No: M 3569).

The immunohistochemical method used in the study was Dako EnVision+™/HRP Mouse, Code K 4001 Dako Denmark A/S. Immunoperoxidase staining was carried out manually.

A semi-quantitative assessment was carried out using a light microscope. Staining reactions were assessed in the cell nuclei (for ER, PgR) and the cell membranes (for HER2). In the case of ER and PgR, the results were shown as:Negative [−], no staining was observed;Positive [+], staining of up to 10% of cells was seen;Strongly positive [++], staining was seen in more than 10% of cells;Very strongly positive [+++], staining was observed in more than 75% of tested structures.

For the purposes of statistical analysis, the results were divided into just two groups,

Negative results [−];Positive results [+,++,+++].

Immunohistochemistry reactions for HER2 were scored by a HercepTest where 0 and 1+ scores are negative, 2+ is weakly positive and 3+ is positive. A positive HER2 result consisted of a uniform and intense membrane staining of more than 30% of tumor cells, and further evaluation was unnecessary for invasive cancers that stained definitely positive or negative. Weakly positive, equivocal or more than two cases were tested for gene amplification by FISH using PATH Vysion TM (Abbott/Vysis: LSI HER2 Spectrum Orange TM and CEP 17 Spectrum-Green TM). A positive result using this method indicated more than six HER2 gene copies per tumor cell nucleus or a HER2 gene to chromosome 17 ratio of more than 2.2 (Table 1).

### 2.3. The Cancer Genome Atlas (TCGA) Database

Breast cancer data from TCGA, which are publicly available from the NCI GDC data portal (https://portal.gdc.cancer.gov/ (accessed on 10 January 2022)), were used in this study as an independent validation set. We performed an in silico analysis to determine the concordance in the frequencies of both studied polymorphisms. The Xena browser was used to analyze overall survival curves in TCGA breast cancer patients (https://xena.ucsc.edu (accessed on 10 January 2022); University of California). Kaplan–Meier plots were presented for the association between ABCB1 polymorphism, individual receptors statuses (ER, PgR, HER2) as well as adjuvant and neoadjuvant chemotherapy. The overall survival analyses of patients with breast cancer treated with chemotherapy relative to 2677 (A) and 3435 (B) ABCB1 polymorphisms were shown. Additionally, for each analysis, patients were divided into two groups: with a presence (VV; red line) or absence of polymorphisms (WV + WW).

### 2.4. Statistical Analysis

Clinical assessment, ABCB1 polymorphism status, ER/PgR/HER2 status and other characteristics were compared according to the follow-up status using the Chi-square statistic. The primary outcome in this analysis was breast cancer survival at 3-, 5-, 7-, 8- and 10-years for all 157 participants. Survival was defined as the survival from the date of diagnosis and subsequent therapy start. We also calculated the odds ratio (OR) and 95% confidence intervals (CIs). A *p*-value of <0.05 was considered statistically significant (Statistica 6.0 and GraphPad Prism 8.0).

## 3. Results

The survival analyses performed for breast cancer patients revealed a similar 10-year survival prognosis when comparing the patients with ABCB1 G2677T/A or C3435T polymorphic variants (VV) versus wild and heterozygous variants (WV and WW) (Figure 1A,B). Similar observations were made for other time intervals, i.e., 3, 5, and 7 (Appendix A). Similarly, no significant difference was observed in the survival rate when different TNM and grading statuses were analyzed relative to polymorphic ABCB1 variants (Appendix A). Importantly, we found a correlation between survival time and individual disease stages, showing the longest survival time in stage I and the shortest survival time in stage III (Figure 1C).

These data were similar to the TCGA breast cancer database (1247 patients), i.e., there was no significant difference in survival between ABCB1 polymorphism carriers (both alleles mutated) and non-carriers (wild-types and heterozygotes) (Figure 2A).

Interestingly, but not surprisingly, further analysis showed an increased survival rate among women with ER-positive and PgR-positive breast cancer relative to both receptors’ negative variants (Figure 1D,E, respectively). Importantly, no such correlation was observed when HER2 status was considered (Figure 1F). Surprisingly, we also found a higher survival chance in the group of individuals that were not subjected to chemotherapy (Figure 1G); however, cancer stage and therapeutic indications must be taken into account in this analysis—in most cases, chemotherapy was enrolled in advanced tumor cases. Moreover, among patients treated with adjuvant chemotherapy, the survival rate was higher compared to neoadjuvant therapy (Figure 1H), but it was not significant (*p* = 0.0791).

We also observed a higher survival rate in ER-positive than in ER-negative cases (Figure 2B). However, in the case of the HER2 status assessment, no significant difference in survival rates was revealed (Figure 2D). A very interesting observation can be made by analyzing survival in PgR status (Figure 2C). The overall survival status in a group of PgR-positive patients is greater in terms of 5-year follow-ups (*p* = 0.04). However, after the next 5 years, that difference is no longer noted. Furthermore, by analyzing adjuvant and neoadjuvant treatment, we can conclude that adjuvant chemotherapy has an advantage, which significantly contributes to a higher survival rate (Figure 2E,F, respectively).

## 4. Discussion

### 4.1. Polymorphisms and Survival Assessment

So far, the role of ABCB1 polymorphisms as prognostic markers remains unclear. The study comprising 914 women did not reveal any association between ABCB1 G2677T/A polymorphism and disease-free survival [25,26]. In contrast, a recent study performed by Li et al. [27] demonstrated that ABCB1 3435TT and ABCG2 421CC were significantly associated with more prolonged survival in Chinese breast cancer patients (100 patients vs. 100 healthy individuals) [27]. In addition, Kim et al. [28] showed that the C3435T polymorphism might be associated with prolonged overall survival and suggested its predictive value in a therapy based on docetaxel [28]. Similar results were obtained by Madrid-Paredes et al. [29], who suggested a prognostic value of C3435T [29].

Another study performed by Green et al. [19] showed that the T/A allele of the ABCB1 G2677T/A gene polymorphism allele was associated with a prolonged disease-free interval (*n* = 53). Similarly, an Australian group comprising 309 women with epithelial ovarian cancer also revealed a prolonged disease-free survival in women carrying the T/A allele [30]. In our study, we revealed no association between those polymorphisms and 10-year survival in breast cancer patients in the Polish population. Consequently, the role of ABCB1 polymorphisms as a prominent factor in predicting treatment response to chemotherapy remains unclear.

### 4.2. Response Prediction

Historically, some studies indicate a significant association between individual ABCB1 polymorphisms (e.g., 1236C > T) and treatment response, but even in that case, no association with treatment outcomes was revealed [16]. Other studies (a meta-analysis) demonstrated that C3435T polymorphism could not be considered a reliable predictor of response to chemotherapy in Caucasian patients with advanced breast cancer [31], which is in concordance with our result. Interestingly, Levy et al. [32] showed an association between C3435T polymorphism and docetaxel pharmacokinetics in breast cancer patients. This could be beneficial in the context of the optimization of an individualized therapy approach. In turn, Paredes et al. [33] demonstrated a lack of association between ABCB1 polymorphisms (C3435T, C1236T, and G2677T/A) and response to anthracyclines-based (doxorubicin and epirubicin) or taxanes-based (paclitaxel and docetaxel) chemotherapy in breast cancer patients.

Nowadays, the best evaluation tools are meta-analyses that suggest (especially those adjusted for other non-genetic factors) that ABCB1 polymorphisms are not associated with genetic susceptibility to breast cancer [34,35]. However, even reports published by the same groups seem a bit confusing when claiming that some of these polymorphisms could be associated with the breast cancer therapy response manifested by the modulation of plasma levels of docetaxel [36]. It brings us to a well-known conclusion that ABCB1 polymorphism status may be one of the numerous factors contributing to the drugs’ metabolism in cancer cells. We also performed the analysis of the survival rate between the groups of (vv) and (ww + wv) patients. However, the evaluated drug response against polymorphism did not show any significant effect (Appendix A).

### 4.3. Receptor Status

Receptor status in cancer cases has always been acknowledged as an essential diagnostic and prognostic factor, especially in breast cancer, where the division into subtypes is mostly based on this element. The three most relevant receptors (ER, PgR, and HER2) basically determine the choice of treatment, which is crucial in the case of advanced tumors. Furthermore, the status of ER, PgR or HER2 is associated with the survival of patients. In our 10-year follow-up study, we collected compelling data confirming that correlation.

Interestingly, we showed an increased survival rate among women with ER- and PgR-positive breast cancer cases that correspond to the results of Poorolajal et al. [37]. Additionally, they showed that breast cancer patients with ER-/HER2+ tumors had shorter survival rates than those bearing ER+/PgR+/HER2- tumors [37]. However, some studies show that the absence of PgR is correlated with prolonged survival [2].

We also showed that ER and PgR expression were good prognostic factors and positively correlated with disease-free survival for up to 10 years. Similar results were shown by [38]. We also showed that neoadjuvant therapy was associated with shorter survival, which is suggested by other authors to result from a decreased expression of ER and PgR receptors caused by this therapy [39]. Our data confirmed that the expression of ER and PgR were good prognostic factors and positively correlated with disease-free long-time survival.

On the other hand, it is known that Herceptin, in combination with neoadjuvant chemotherapy applied in HER2 positive tumors, is associated with a reduced risk of recurrence and improvement of cancer-free survival [37]. Importantly, there was no difference in time survival when HER2 status was considered (both our and TCGA data) in our work, which was the opposite of the results demonstrated by Slamon et al. [40]. However, the usefulness of this parameter (HER2 status) cannot be estimated since different cancer characteristics can require different therapy regimens, which leads to a different response in individual cases.

## 5. Summary and Conclusions

We still do not have a clear answer concerning the level of contribution of ABCB1 genetic variants to cancer risk or therapy response. Different studies show diverse results, which may result from the fact that these mechanisms are too complex to refer to single SNPs nucleotides. One of the limiting factors is the lack of prospective studies that would include “monitoring” of individuals for a longer period and would include genetic background and exposition (time and dose) to a vast number of risk factors. Similarly, such studies should include comprehensive population studies that may become available as the new high throughput technologies are developed.

Altogether it seems that the status of the ABCB1 polymorphism may be one of the numerous factors that affect the metabolism of cancer (development and therapy response) but without a leading role in this phenomenon. Consequently, it may not be treated as a reliable prognostic or predictive marker in assessing breast cancer patients’ survival or the response to therapy of individual tumor types or characteristics. However, there is no doubt that ABC family proteins play a pivotal role in the metabolism of numerous drugs and should be assessed (activity, quantity, and quality) to obtain a better perspective for cancer patients. It may be that novel molecular biology methods like RNAseq or GWAS may change the perspective and contribute to more unambiguous conclusions.

## Figures and Tables

**Figure 1 genes-13-00729-f001:**
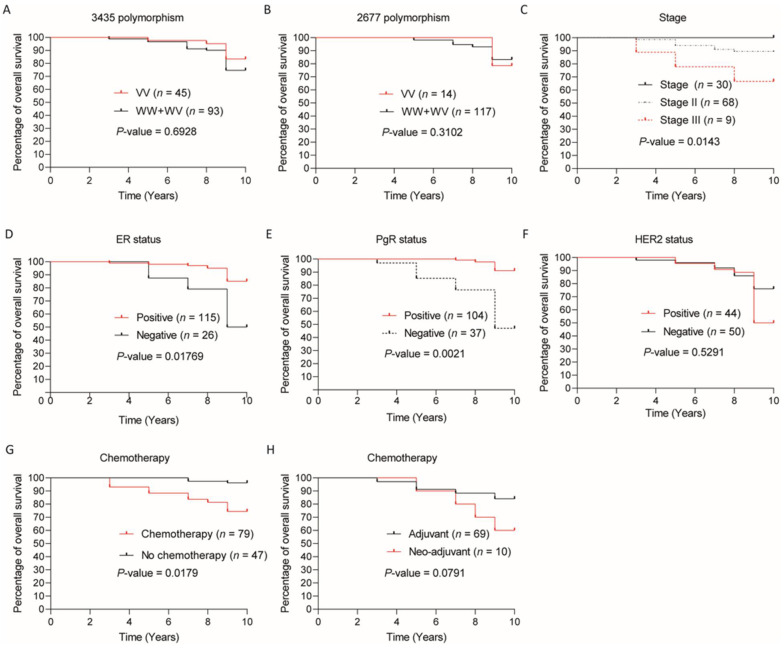
Ten-year survival assessment of breast cancer patients. Kaplan–Meier curves of overall survival of patients with breast cancer for ABCB1 3435 (**A**) and 2677 (**B**) polymorphisms. For each analysis, patients were divided into two groups: with a presence (VV; red line) or absence/heterozygous variant of polymorphisms (WV + WW; black line). (**C**) Kaplan–Meier curves of overall survival of patients with breast cancer in different stages of disease (Stage I—black line; Stage II—green line; Stage III—red line). Kaplan–Meier curves of the overall survival of patients with breast cancer with ER (**D**), PgR (**E**), and HER2 (**F**) receptor status were analyzed; red line—positive, black line—negative. Impact of chemotherapy treatment ((**G**) black line—patients treated with chemotherapy; red line—patients non-treated with chemotherapy) and type of chemotherapy ((**H**) black line—adjuvant chemotherapy; red line—neoadjuvant chemotherapy) on overall survival was also assessed. All Kaplan–Meier curves were generated using GraphPad Prism (data collected from Xena browser).

**Figure 2 genes-13-00729-f002:**
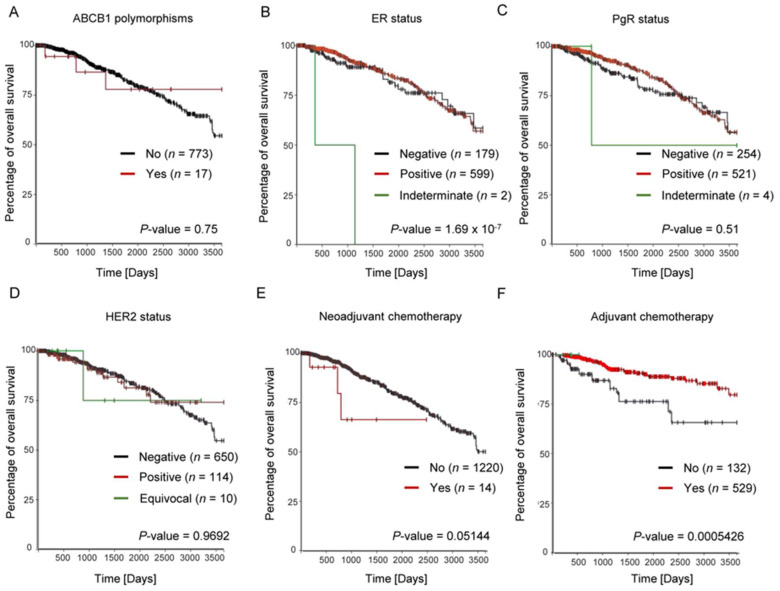
Overall survival analysis in reference to the TCGA breast cancer database. (**A**) Kaplan–Meier curves of overall survival of patients with breast cancer for ABCB1 polymorphisms. For each analysis, patients were divided into two groups: with a presence (red line) or absence of polymorphisms (black line); Kaplan–Meier curves of overall survival of patients with breast cancer with ER (**B**), PgR (**C**), and HER2 (**D**) receptor status was analyzed; red line—positive, black line—negative. The impact of neoadjuvant chemotherapy treatment ((**E**) black line—treated patients; red line—non-treated patients) and adjuvant chemotherapy ((**F**) black line—treated patients; red line—non-treated patients) on overall survival was also assessed. All Kaplan–Meier survival curves were generated using the Xena browser.

**Table 1 genes-13-00729-t001:** Clinical characteristics of study participants. “Missing” stands for the patients who, relative to the study from 2012, were impossible to contact or the parameter was not found in the database.

Characteristic	No. of Patients	% of Patients
3435 polymorphism		
VV	45	28.7
WV	60	38.2
WW	33	21
Missing	19	12.1
2677 polymorphism		
VV	14	8.9
WV	57	36.3
WW	60	38.2
Missing	26	16.6
Grading status		
G1	46	29.3
G2	59	37.6
G3	32	20.4
Gx	5	3.2
Missing	15	9.5
T status		
T1	65	41.4
T2	58	36.9
T3	4	2.5
T4	3	1.9
Tx	3	1.9
Tis	1	0.6
Missing	23	14.6
N status		
N0	50	31.8
N1	74	47.1
N2	7	4.5
N3	1	0.6
Nx	2	1.3
Missing	23	14.6
M status		
M0	134	85.4
Mx	2	1.3
Missing	21	13.4
Stage		
I	30	19.1
II	68	43.3
III	9	5.7
Missing	50	31.9
PgR status		
“+”	104	66.2
“−”	37	23.6
Missing	16	10.2
ER status		
“+”	115	73.2
“−”	26	16.6
Missing	16	10.2
HER2 status		
“+”	44	28
“−”	50	31.8
Missing	63	40.2
Chemotherapy		
Adjuvant (AC or CMF)	69	87.3
Neoadjuvant (AC or CMF)	10	12.7
Yes	79	50.4
No	47	29.9
Missing	31	19.7

## Data Availability

All relevant data are included in the manuscript.

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
