# Peer review of "No Association between ABCB1 G2677T/A or C3435T Polymorphisms and Survival of Breast Cancer Patients—A 10-Year Follow-Up Study in the Polish Population"

_genes, 2022, doi:10.3390/genes13050729_

Round 1
Reviewer 1 Report
The authors in this study evaluated the association between the polymorphisms in the genes coding for ABC family proteins and the survival of breast cancer patients. Further, they assessed the estrogen, progesterone, and human epidermal growth receptors status on the survival of cancer patients. For the above analyses, they used breast cancer patients’ data in Poznan University of Medical Sciences in Poznan, Poland, and TCGA data. Based on the above analyses, they found that the status of ABCB1 polymorphism does not have a strong correlation to patients’ survival. However, they found that ER and PgR status significantly affect patients’ survival.
The manuscript is written clearly and well organized. Few things can be addressed and incorporated to strengthen their study.
Major:
- From the breast cancer patients’ data, can you graphically show the survival rate between patients having polymorphism (vv) and non-polymorphism (ww+wv) and who had chemotherapy treatment. In other words, figure 1G shows the survival of all patients who had chemotherapy treatment. Instead of all patients, separate the survival rate in response to chemotherapy of patients with the presence of polymorphism (vv) and without the polymorphism (ww+wv). If you can show this, the drug response against polymorphism could be evaluated. Once you demonstrate these data as figures, please discuss them in the manuscript accordingly.
Minor:
- Line 69: Please write out the full name for PgP.
Author Response
We want to thank the reviewer for a thorough revision and constructive comments. Please find the response below. All changes in the text were made in Truck Changes mode.

Reviewer 2 Report
Overall comment:
The hypothesis is good and raises critical clinical issues. However, background information is superficial and did not give sufficient insight into the hypothesis.
The materials and methods section were not comprehensive. Authors should incorporate more information associated with this manuscript.
One of the major issues in the present state of the manuscript is scientific language and formatting. Non-specific terminology and overstatement is the major issue of the manuscript. The present state of the manuscript required a significant amount of revision to meet publication standards.
Suggestion: the title should be “No association between ABCB1 G2677T/A or C3435T Polymorphisms and Survival of Breast Cancer Patients – a 10-Year Follow-up Study in the Polish population”
Major comments:
- Examples of non-specific terminology, scientific language and formatting errors;
- Page 1, Line 38; “Statistics show that the main cause for cancer therapy failure is resistance to therapeutic agents.“ Is it statistics or statistical analysis? Also, statistical analysis may provide a significant association between variables.
- Line 76-78; “However, there are some contradictory data concerning the possibility of using ABCB1 polymorphisms as reliable risk factors or predictors of response to chemotherapy in breast cancer patients”. What is the meaning of “there are some contradictory data”? Authors should specifically cite previously described data.
- Throughout the manuscript, numerous non-specific sentences represent serious issues. Authors should scientifically proofread.
- The background and discussion section should be referenced heavily. Example; sentence 38-42, no specific literature cited, and ditto throughout MS.
- Why 10-year follow up? What about 5-year follow up results? Hypothesis lack insight. Authors should precisely define the hypothesis by incorporating specific literature in the manuscript.
- The materials and methods section is a real “disaster”. What are the patient selection criteria? Why were only 157 patients selected? What about histopathology information? ER/PR/HER2 positivity (histoscore) methodology? Etc.......All information associated with Table 1 should be incorporated in the manuscript as supplementary information.
- Line 113-114: “Importantly, some data are missing due to different issues appearing over the time including decease of some patients, loss of contact, distant relocation.” What is the meaning of some data? Some patients?? Etc.....All details should be incorporated in the manuscript.
- Also, detailed information on TCGA datasets should be incorporated in the manuscript. And why compared to TCGA datasets?? What is the clinical relevance? What is the difference or similarity between patient clinicopathological parameters? Etc...Such information is essential.
- All associated data of breast cancer survival at 3-, 5-, 7-, 8- and 10-years should be incorporated in the manuscript.
- All jargon such as “some”, “good”, these, many, intensive, historically, different studies, etc...should be excluded. Proofread !!!!!!
- Concluding remarks are superficial.
- Gene nomenclature should be consistent, eg, PR or PgR.
- A significant amount of proofreading and revision is needed. For every section of the manuscript.
Author Response

(The authors gave the same response as above.)

Round 2
Reviewer 1 Report
The comments are addressed adequately and the manuscript quality is improved. Minor formatting may be required. Please check the spacing and indents between paragraphs.
Reviewer 2 Report
The current state of the revised manuscript substantially improved. Final proofreading is recommended.